# COVID-19 and Inflammatory Bowel Disease: Patient Knowledge and Perceptions in a Single Center Survey

**DOI:** 10.3390/medicina56080407

**Published:** 2020-08-13

**Authors:** Rocco Spagnuolo, Tiziana Larussa, Chiara Iannelli, Cristina Cosco, Eleonora Nisticò, Elena Manduci, Amalia Bruno, Luigi Boccuto, Ludovico Abenavoli, Francesco Luzza, Patrizia Doldo

**Affiliations:** 1Clinical and Experimental Medicine Department, “Magna Graecia” University, Viale Europa, 88100 Catanzaro, Italy; cosco.cristina@libero.it (C.C.); eleonoranis7@gmail.com (E.N.); manducielena93@gmail.com (E.M.); amalia.bru93@yahoo.it (A.B.); doldo@unicz.it (P.D.); 2Health Sciences Department, “Magna Graecia” University, Viale Europa, 88100 Catanzaro, Italy; tiziana.larussa@gmail.com (T.L.); iannelly@yahoo.it (C.I.); l.abenavoli@unicz.it (L.A.); luzza@unicz.it (F.L.); 3Greenwood Genetic Center, Clemson University, Clemson, SC 29631, USA; lboccuto@ggc.org

**Keywords:** COVID-19, inflammatory bowel disease, knowledge, anxiety, pandemic, health care, prevention measures

## Abstract

*Background and objectives*: Spreading of SARS-CoV-2 infection from China to countries with a higher prevalence of inflammatory bowel disease (IBD) has generated concern among gastroenterologists and patients. The aim of this survey is to evaluate knowledge about clinical importance of COVID-19, disease management, prevention measures, and anxiety level during pandemic among patients with IBD. *Material and methods*: From 15th March to 15th April 2020, a questionnaire survey was administered to 200 patients with IBD by email or phone application. The questionnaire consisted of five sections: (1) anthropometric, demographic and clinical characteristics, (2) knowledge about clinical importance of COVID-19, (3) IBD management, (4) prevention measures, (5) anxiety level during pandemic. *Results*: One hundred forty two questionnaires were completed. Ninety-seven patients (68.3%) were males with a mean age of 46 years (SD 13; range 17–76)**.** Fifty-four individuals (38%) were affected by Crohn disease and 88 (62%) by Ulcerative Colitis. Most patients reported high knowledge about clinical importance of COVID-19 (80%), IBD management (72%), and prevention measures (97%). Sixty-two percent of them showed moderate-high level of anxiety. High education level was independently associated with high knowledge about clinical importance of COVID-19 (odds ratio [OR] 5, 95% confidence interval [CI] 1.49–16.6, *p * = 0.009) and older age (OR 1, 95%, CI 1.01–1.1, *p* = 0.01), while the receipt of e-format educational material with low knowledge about clinical importance of COVID-19 (OR 3, 95%, CI 1.08–9.3, *p* = 0.03). Displaying an active disease appeared to be independently associated with low knowledge of IBD management (OR 5.8, 95% CI 1.4–22.8, *p* = 0.01) and no variables other than an older age was independently associated with higher level of anxiety (OR 1.04, 95% CI 1.009–1.09, *p* = 0.01). *Conclusions*: High educational level and aging promote knowledge about clinical importance of COVID-19, while e-format educational material does not. Taken together with findings that an active disease status compromises knowledge of IBD management and the high level of anxiety related to increasing age, these data suggest the need of further supporting patient-oriented strategies in IBD during Covid-19 pandemic.

## 1. Introduction

Following the first reports of cases of acute respiratory syndrome in the Chinese Wuhan municipality at the end of December 2019, Chinese authorities have identified a novel coronavirus as the main causative agent. On 12 February 2020, the novel coronavirus was named severe acute respiratory syndrome coronavirus 2 (SARS-CoV-2) while the disease associated with it is now referred to as COVID-19 [1]. On 11 March 2020, World Health Organization stated that the outbreak might be uncontrolled and made the assessment that COVID-19 can be characterized as a pandemic. As of 17 June 2020, 8,061,550 cases of COVID-19 were reported worldwide in more than 177 countries and regions, with 440,290 total confirmed deaths. Considering the ratio of individuals tested positive to the COVID-19 over population, Italy is the fourth worst-affected European country in the pandemic, with 237,500 cases and 34,405 deaths [2]. However, these data are constantly updated on international data resources. From the available evidence, COVID-19 infection causes a mild disease (i.e., no pneumonia or mild pneumonia) in about 80% of cases and in most cases it resolves. About 14% have a more serious illness and 6% have a critical illness [3]. The vast majority of the most serious diseases and deaths have occurred among the elderly individuals and those with pre-existing chronic conditions.

Patients with inflammatory bowel disease (IBD) are at increased infection risk, especially when being treated with steroids, immune-suppressants or biologics. The nature and magnitude of this risk vary with the type of immunosuppressive drug and with the patient’s sex and age [4]. The overall available evidence suggests that patients with IBD do not have an increased risk of developing COVID-19 and should stay on IBD medications. Patients receiving immune-suppressors should be carefully monitored for the occurrence of symptoms and/or signs suggesting COVID-19 [5].

Spreading of the outbreak from China to countries with a higher prevalence of IBD has generated concern among gastroenterologists and patients, however, a recent systematic review has shown that patients with IBD do not appear to be more susceptible to SARS-CoV-2 infection and there is no evidence of an association between IBD therapies and increased risk of COVID-19. IBD medication adherence should be encouraged to prevent disease flare but, wherever possible, high dose systemic corticosteroids should be avoided [6]. At the time of writing, there are 1511 patients with COVID-19 and IBD worldwide. The countries with the most reported cases are US, UK, and Spain. In Italy about 79 cases have been reported showing active IBD, old age and comorbidities were associated with a negative COVID-19 outcome, whereas IBD treatments were not [7].

Only a recent survey online on European Federation of Crohn’s & Ulcerative Colitis Associations (EFCCA) [8] yet not published and conducted on 3815 patients, reported concern and fear of patients about management of IBD and normal daily activities, during pandemic.

Anxiety level in patients affected by chronic immune-mediated inflammatory diseases (IMIDs) has been widely quantified through several patients reported outcomes showing higher level in comparison with general population [9]. Actually, various studies evaluated the anxiety level in general population during pandemic. A Cross-Sectional Survey on the Psychological Impact of the COVID-19 on patients with IBD has shown that about half (48%) expressed symptoms of anxiety [10].

The aim of this study is to perform a survey about knowledge about clinical importance of COVID-19, knowledge of disease management, prevention measures, and anxiety level during the pandemic among patients with IBD.

## 2. Materials and Methods

### 2.1. Study Cohort

From 15 March to 15 April 2020, a questionnaire survey was administered to patients with IBD by email or phone application. Patients were affected by Crohn’s disease (CD) and Ulcerative Colitis (UC) and were followed by Gastroenterology and Digestive Physiopathology Unit of “Mater Domini” University Hospital. Diagnosis of CD or UC was established according to standard clinical, endoscopic, histological, and radiological criteria [11,12]

### 2.2. Instruments

The questionnaire was defined and organized into five sections, dealing with: (1) demographic and clinical characteristic, (2) knowledge about clinical importance of COVID-19, (3) knowledge about management of IBD, (4) knowledge of prevention measures, (5) anxiety level during pandemic, respectively.

First section was composed by anthropometric, demographic, and clinical data of patients, including age, gender, educational level defined as having a high school diploma, marital status, and parental status, number of cohabitants, employment, type of occupation and workplace, use of smartphone or personal computer (pc) as source of COVID-19 information, reception of educational material by gastroenterologist to inform patients about management of IBD during COVID-19 emergency and, finally, any presence of a family member with COVID-19. Clinical data included disease type, duration and prescribed treatment. Disease activity was measured by Harvey Bradshaw index for CD [13] and partial Mayo score for UC [14]. Furthermore, disease activity was dichotomized in “low” with a total score below 4 for partial Mayo score and below 7 for Harvey Bradshaw index, “high” with a score above 5 for partial Mayo score and above 8 for Harvey Bradshaw index.

Finally, eventual diagnoses of psychiatric diseases and other chronic diseases were recorded. In the second section, as shown in previous studies [15], 12 questions were used to collect information regarding knowledge about clinical importance of COVID-19. Each answer was graded as 0 (incorrect answers) and 1 (correct answers). Maximum score was 12 and minimum 0. The score was dichotomized in “low degree” when ≤6 and “high degree” when >6.

In the third section, 10 questions were used to collect information regarding patients’ knowledge about management of IBD during pandemic. Each answer was graded as 0 (incorrect answers) and 1 (correct answers). Maximum score was 10 and minimum 0. The score was dichotomized in “low degree” of knowledge when ≤5 and “high degree” of knowledge when >5.

In the fourth section, as shown in previous studies [16], 10 questions were used to collect information regarding patients’ knowledge of prevention measures during pandemic. Each answer was graded as 0 (incorrect answers) and 1 (correct answers). Maximum score was 10 and minimum 0. The score was dichotomized in “low degree” of knowledge when ≤5 and “high degree” of knowledge when >5.

Fifth section evaluated anxiety level during COVID-19 emergency by State-Trait Anxiety Inventory (STAI) [17]: a Self-completed questionnaire of 40 items which aims to assess separately the state of anxiety (defined as temporary and influenced by the contingent situation in which the respondent notes how he feels right now) and the anxiety trait (propensity to be anxious where the respondent see how it feels “generally”) with 20 items each. The total score ranges from 0–160; score less than 41 was classified as no anxiety, 41–80 mild anxiety, 81–120 moderate anxiety and 121–160 severe anxiety. Furthermore, anxiety levels were dichotomized in “low anxiety” with score less than 81 and “moderate-high” with score higher than 80.

### 2.3. Statistical Analysis

Data were described using means and standard deviation, numbers, and proportions as appropriate. Univariate analysis was performed by chi-square test and independent sample *t* test to analyze factors influencing outcomes of interest. A multivariate analysis was performed by logistic regression models adjusting for type of IBD, gender, age, educational level, marital status, parental status, job status, use of smartphone/pc, educational material received by gastroenterologist, disease duration biological therapy active disease, anxiety. The odds ratio (OR) was used as a measure of association between the results of the questionnaires and the presence of a certain variable.

All statistical analyses were performed using the Statistical Package for Social Sciences (version 19.0; SPSS, Inc., Chicago, IL, USA). *p* values < 0.05 were considered as statistically significant.

### 2.4. Ethics

This study was approved by the local ethics committee of Magna Graecia University on 15 March 2018 (protocol number 179/2020). This study was conducted in compliance with the principles outlined in the Declaration of Helsinki. Informed written consent was sent together with the questionnaire and obtained from each participating patient.

## 3. Results

### 3.1. Background of Study Population

A total of 200 questionnaire were sent. Out of these, 142 were completed, with a response rate of 71%. Detailed anthropometric, demographic and clinical characteristics of the patients are shown in Table 1.

### 3.2. Knowledge about Clinical Importance of COVID-19

One hundred fourteen patients (80%) have shown high degree knowledge about clinical importance of COVID-19, while an older age has been found independently associated with it (OR 1.06, 95% CI: confidence interval 1.01–1.1, *p* = 0.01) (Table 2).

Most patients (91.5%) reported that COVID-19 is a viral infection, originated from Asia (94%), with human-to-human transmission mainly via droplets (87%) and an incubation period between 2 and 14 days (70%). Moreover, almost all (99%) the patients with IBD correctly answered about clinical manifestations of COVID-19, indicating fever, cough, and dyspnea as the most common symptoms while reporting (92%) that oro-nasopharyngeal swab was used for diagnosis of COVID-19. Participants were also informed about the lack of effective therapy (77%) or available vaccine (98%), and they correctly reported also that elderly people were at increased risk of infection (67%).

Patients with high knowledge about clinical importance of COVID-19 reported high educational level more often than patients with low knowledge about clinical importance of COVID-19 (72% vs. 43%, *p* = 0.04, respectively). This was consistent with the highlight that high education level was independently associated with high knowledge about clinical importance of COVID-19 (OR 5, 95% CI 1.49–16.6, *p* = 0.009) (Table 2).

The receipt of informative material by gastroenterologist was more frequent in patients with low knowledge about clinical importance of COVID-19 compared to those with high knowledge about clinical importance of COVID-19 (78% vs. 54%, *p* = 0.01, respectively) (Table 2). Accordingly, receiving educational material on COVID-19 by gastroenterologist was independently associated with low knowledge about clinical importance of COVID-19 (OR 3, 95% CI 1.08–9.3, *p* = 0.03) (Table 2). Furthermore moderate-high level of anxiety was independently associated with low level of knowledge (OR 5, 95% CI 1.49–16.6, *p* = 0.009) (Table 2).

### 3.3. Knowledge of IBD during Pandemic

Most participants (72%) reported a high knowledge concerning the management of IBD. Just over half (51%) declared that their intestinal disease could increase the risk of infection with SARS-CoV-2 and much more (72%) that it could worsen the outcome of COVID-19 infection. Although almost all patients (96%) considered appropriate to continue their current medical therapy, fewer (60%) believed that the treatment does not increase the risk of infection with SARS-CoV-2. Most of the patients (68%) agreed on the remodulation of hospital activity during COVID-19 pandemic, with suspension of routinely visits and almost all (95%) considered appropriate going to the hospital only in case of clinical relapse. An active disease status was independently associated with a low knowledge of IBD management during pandemic (OR 5.8, 95% CI 1.4–22.8, *p* = 0.01) (Table 3). Nevertheless, the remaining variables were not associated with the level of knowledge of IBD management during pandemic.

### 3.4. Knowledge of Prevention Measures during Pandemic

Almost all patients with IBD (97%) presented a good knowledge of prevention measures such as utility of frequent hand washing (100%), use of disinfectant (99%), use of masks (94%), and importance of social distancing (99%).

### 3.5. Anxiety Level during Pandemic

Eighty-eight (62%) participants reported a moderate-high level of anxiety. Older age was independently associated with moderate-high level of anxiety (OR 1.04, 95% CI 1.009–1.09, *p* = 0.01). More patients with a low level of anxiety were employed (72% vs. 57%, *p* = 0.04) and used smartphone or pc as a source of COVID-19 information (93% vs. 82%, *p* = 0.05). However, these associations disappeared when adjusted for the other variables. At the same time, no other variable was associated with the level of anxiety (Table 4).

## 4. Discussion

In the current context of COVID-19 pandemic, IBD units have been forced to dramatically change and restructure management of patients with IBD [18]. This study aimed to investigate the level of knowledge about clinical importance of COVID-19 and perceptions among patients suffering from IBD.

As expected, a high knowledge about clinical importance of COVID-19 was found to be associated with high educational level. Consistently, a previous study [15] among Chinese residents demonstrated a strong association between degree of information about COVID-19 and higher educational level. Nevertheless, this is the first study dealing with the knowledge about clinical importance of COVID-19 in a cohort of patients with IBD.

From the beginning of the pandemic in Italy, several IBD units attempted to improve remote contact with patients providing them with educational material containing relevant information about COVID-19 and IBD [19].

Surprisingly, in this study, patients who had received this type of information displayed a lower knowledge about clinical importance of COVID-19 compared to those who had not, and this was confirmed to be an independent association. A possible explanation could be that the choice of a digital format was not the preferred way in our cohort of patients. Likewise, this suggest that doctor–patient interpersonal interaction is an unquestionably important feature in the health system in order to improve knowledge, health-related behavior change, and self-management of patients with IBD.

The finding that a higher level of anxiety was independently associated with a lower level of knowledge about clinical importance of COVID-19 was not surprising. That is, anxiety significantly impairs patient performance in getting the right available information in the field. As a consequence, gastroenterologists should take into account the high level of anxiety in the context of COVID-19 pandemic to further support the related needs of the patient with IBD.

Furthermore, it was not surprising that increasing age was independently associated with high level of knowledge about clinical importance of COVID-19. Indeed, older patients are more prone than younger to pay attention on health-related issues which normally increase with aging.

At date, several ongoing studies evaluate management of IBD patients during pandemic reporting as CD and UC does not increase risk of infection [6,7] and of severe forms of COVID-19 [20,21]. Moreover, the actual risk of infection or of development of COVID-19 due to medical therapy is unknown [22]. Long-term data are needed to define the best strategy, considering that stopping therapy exposes patients to a greater risk of disease recurrence and therefore this decision should be individualized [23,24]. All European hospitals intensively reduced routine activities to prepare for high numbers of admissions of patients with COVID-19 and there was a readjustment of care and resetting of clinical priorities [25]. However, patients with IBD should be reminded that this interruption is temporary and alternative solutions have been found, including remote monitoring, drug home delivery, and limitations for infusion units [19].

As far as we know, no validated instruments have been developed in order to measure patients’ knowledge of IBD management during COVID-19 pandemic. Therefore, we built a questionnaire according to the available published data [15,16]. The high rate (72%) of patient who declared to have high level of knowledge of IBD management during pandemic is probably due to the widespread information by the medical system and IBD Units as well as the several media channels and forum they certainly attended in that period. Nevertheless, the sole variable that showed to be independently associated with a low knowledge of IBD management resulted to be the activity status of the disease. This is probably due to the fact that the deeper patients are involved in their disease (i.e., active disease), the lower is their knowledge about how properly manage it. Once again, consistently with previous study [26], this finding suggests that gastroenterologists have to pay more attention at patients with IBD in active disease status, further addressing the knowledge of how to manage IBD during pandemic in this setting of patients.

Several studies have been published about psychological impact of the outbreak in the general population and demonstrated an increase of anxiety level during pandemic. Furthermore, many patients seeking medical consultation for gastrointestinal problems show an associated affective disorder [27]. Patients with IBD are 3–5 times more likely to develop anxiety disorders [28,29,30].

According with recent study [10] we confirmed that IBD patients have a high level of anxiety during Covid-19 pandemic. However, no variable other than an increasing age has been shown to be independently associated with this findings. It is not surprising that mood disorders are more frequent in older than younger patients.

Even thought to be employed and the use of smartphone/pc have been shown to be associated with lower level of anxiety, they both still not maintained statistical significance when adjusted for the other variables.

Limitation of the study was constituted mainly by the relatively small sample size, because of the fact that data collection was performed exclusively through email as routinely visits were suspended. The non-prospective evaluation of patients did not allow us to highlight the impact of the pandemic on disease management, but this was beyond the scope of this survey.

## 5. Conclusions

In conclusion, this study shows that in patients with IBD a high educational level promotes knowledge about clinical importance of COVID-19 while an active disease status lowers the knowledge of IBD management. Taken together with the high level of anxiety and its relationship with increasing age, this suggests that gastroenterologists have to further support the needs of patients with IBD with targeted strategies during COVID-19 pandemic.

## Figures and Tables

**Table 1 medicina-56-00407-t001:** Demographic, anthropometric, and clinical characteristics of the 144 patients with inflammatory bowel disease.

	UC (88)	CD (54)	TOT (144)
Male gender	60 (62)	37 (38)	97 (68)
Age, years	47 ± 13	46 ± 14	46 ± 13
High educational level	61 (65)	33(35)	94(66)
Marital status	59 (67)	29 (33)	88 (61)
Parental status	63 (69)	29 (31)	92 (65)
Number of cohabitants	66 (64)	37 (36)	103 (72)
Employed	55 (64)	31 (36)	86 (60)
Working outside home	29 (57)	22 (43)	51 (36)
Use of smartphone/pc as source of COVID-19 information	78 (64)	44 (36)	122 (86)
Receipt of educational material about COVID-19 by gastroenterologist	55 (65)	29 (35)	84 (59)
Disease’s characteristics			
Disease duration, years	8 ± 7	11 ± 8	9 ± 8
Active Disease	4 (31)	9 (69)	13 (9)
Harvey Bradshaw index score	5 ± 3		
Partial Mayo Score		3 ± 2	
Treatment			
Mesalamine	72 (68)	34 (32)	106 (74)
Steroids	4 (67)	2 (33)	6 (4)
Thiopurines	1 (50)	1 (50)	2 (1)
Biologics	26 (54)	22 (46)	48 (34)

Continuous variables are shown as mean ± standard deviation (SD), categorical variables are presented as number and proportion (%). Abbreviations: UC: Ulcerative Colitis; CD: Crohn Disease; TOT: total.

**Table 2 medicina-56-00407-t002:** Characteristics of the 142 patients with inflammatory bowel disease according to the level of knowledge about clinical importance of COVID-19.

	Knowledge of COVID-19				
	Low *n* = 28	High *n* = 114	*p*	OR (95% CI)	*p* (MVA)
Gender, *n* (%) Female Male	10 (35) 18 (65)	35 (31) 79 (69)	0.3	0.6 (0.2–1.7)	0.3
Age, years (mean ± SD)	46 ± 14	47 ± 13	0.6	1.06 (1.01–1.1)	**0.01**
High educational level, *n* (%) Yes No	12(43) 16 (57)	82 (72) 32 (28)	**0.04**	5 (1.49–16.6)	**0.009**
Marital status, *n* (%) Yes No	8 (28) 20 (72)	46 (40) 68 (60)	0.1	2.6 (0.6–11)	0.1
Employed, *n* (%)	16 (58)	73 (65)	0.3	1.2 (0.4–3.4)	0.3
Use of smartphone/pc, *n* (%)	21(75)	101 (89)	0.06	0.4 (0.1–2.2)	0.3
Educational material, *n* (%) Yes No	22 (78) 6 (22)	62 (54) 52 (46)	**0.01**	3.1 (1.08–9.3)	**0.03**
Type of IBD, *n* (%) Crohn’s disease Ulcerative Colitis	7 (25) 21 (75)	62 (54) 52 (46)	0.08	2.4 (0.7–7.6)	0.1
Disease duration, years (mean ± SD)	8 ± 7	9 ± 7	0.9	1.04 (0.3–3.4)	0.9
Biological therapy, *n* (%) Yes No	8 (28) 20 (72)	40 (35) 74 (65)	0.3	0.8 (0.2–2.7)	0.8
Active disease, *n* (%) Yes No	2 (8) 26 (92)	11 (10) 103 (90)	0.5	0.6 (0.1–4.3)	0.6
Anxiety, *n* (%) Moderate-high Low	21 (75) 7 (25)	67 (58) 47 (42)	0.08	5 (1.1–11)	**0.03**

Continuous variables are shown as mean ± standard deviation. Categorical variables are presented as number and proportion. *p* values were calculated by independent sample *t* test for continuous variables and chi-square for categorical variables (*p* < 0.05). Odds ratios and confidence intervals were calculated using multivariate logistic regression model. *p* values < 0.05 are reported in bold. Abbreviations: IBD: Inflammatory Bowel Disease; SD: standard deviation; OR: odds ratio; CI: confidence interval; MVA: multivariate analysis.

**Table 3 medicina-56-00407-t003:** Characteristics of the 142 patients with inflammatory bowel disease according to their knowledge of IBD management during COVID-19 pandemic.

	Knowledge of IBD Management				
	Low *n* = 39	High *n* = 103	*p*	OR (95% CI)	*p* (MVA)
Gender, *n* (%) Female Male	14 (36) 25 (64)	31 (30) 72 (70)	0.3	0.8 (0.3–2)	0.6
Age, (mean) years± SD	47 ± 13	46 ± 13	0.6	0.9 (0.9–1.01)	0.2
High educational level, *n* (%) Yes No	25 (64) 14 (36)	69 (67) 34 (33)	0.4	0.6 (0.2–1.8)	0.4
Marital status, *n* (%) Yes No	16 (41) 23 (59)	38 (37) 65 (63)	0.3	1.2 (0.4–3.7)	0.6
Parental status, *n* (%) Yes No	23 (59) 16 (41)	69 (67) 34 (33)	0.2	0.4 (0.1–1.2)	0.1
Employed, *n* (%)	25 (64)	64 (62)	0.5	0.7 (0.3–1.8)	0.5
Use of smartphone/pc, *n* (%)	33 (85)	89 (86)	0.4	1.6 (0.3–6.8)	0.5
Educational material, *n* (%) Yes No	21 (54) 18 (46)	63 (61) 40 (38)	0.2	0.6 (0.3–1.5)	0.3
Type of IBD, *n* (%) Crohn’s disease Ulcerative Colitis	14 (36) 25 (64)	40 (39) 63 (61)	0.4	1.6 (0.6–4.1)	0.2
Disease duration, years (mean ± SD)	9 ± 8	9 ± 7	0.5	0.8 (0.3–2.06)	0.6
Biological therapy, *n* (%) Yes No	13 (34) 26 (66)	35 (34) 68 (66)	0.5	1.2 (0.5–3.01)	0.6
Active disease, *n* (%) Yes No	7 (18) 32 (82)	6 (6) 97 (94)	**0.03**	5.8 (1.4–22.8)	**0.01**
Anxiety, *n* (%) Moderate-High Low	26 (66) 13 (34)	62 (60) 41 (40)	0.3	0.7 (0.3–1.8)	0.5

Continuous variables are shown as mean ± standard deviation. Categorical variables are presented as number and proportion. *p* values were calculated by independent sample *t* test for continuous variables and chi-square for categorical variables (*p* < 0.05). Odds ratios and confidence intervals were calculated using multivariate logistic regression model. *p* values < 0.05 are reported in bold. Abbreviations: SD: standard deviation; OR: odds ratio; CI: confidence interval; MVA: multivariate analysis.

**Table 4 medicina-56-00407-t004:** Characteristics of the 142 patients with inflammatory bowel disease according to the level of anxiety.

	Anxiety Level				
	Low *n* = 54	Moderate/High *n* = 88	*p*	OR (95% CI)	*p* (MVA)
Gender, *n* (%) Female Male	15 (28) 39 (72)	30 (34) 58 (66)	0.2	1.5 (0.6–3.6)	0.3
Age, (mean) years ± SD	43 ± 13	49 ± 13	0.7	1.04 (1.009–1.09)	**0.015**
High educational level, *n* (%) Yes No	39 (72) 15 (28)	55 (63) 33 (37)	0.1	0.9 (0.3–2.5)	0.9
Marital status, *n* (%) Yes No	19 (35) 35 (65)	35 (40) 53 (60)	0.3	2.5 (0.8–7.4)	0.09
Parental status, *n* (%) Yes No	33 (61) 21 (39)	59 (67) 29 (33)	0.2	0.7 (0.2–2.2)	0.6
Employed, *n* (%)	39 (72)	50 (57)	**0.04**	0.5 (0.2–1.3)	0.2
Use of smartphone/PC, *n* (%)	50 (93)	72 (82)	0.05	1.2 (0.2–5.5)	0.7
Educational material, *n* (%) Yes No	35 (65) 19 (35)	49 (56) 39 (44)	0.1	1.6 (0.7–3.6)	0.1
Type of IBD, *n* (%) Crohn’s disease Ulcerative Colitis	21 (39) 33 (61)	33 (37) 55 (63)	0.5	1.1 (0.4–2.5)	0.7
Disease duration, years (mean ± SD)	9 ± 7	9 ± 7	0.7	1.7 (0.7–4.3)	0.2
Biological therapy, *n* (%) Yes No	18 (34) 36 (66)	30 (34) 58 (66)	0.5	0.7 (0.2–1.6)	0.4
Active disease, *n* (%) Yes No	5 (9) 49 (91)	8 (9) 80 (91)	0.6	1.1 (0.2–5.09)	0.8

Continuous variables are shown as mean ± standard deviation. Categorical variables are presented as number and proportion. *p* values were calculated by independent sample *t* test for continuous variables and chi-square for categorical variables (*p* < 0.05). Odds ratios and confidence intervals were calculated using multivariate logistic regression model. *p* values < 0.05 are reported in bold. Abbreviations: SD: standard deviation; OR: odds ratio; CI: confidence interval; MVA: multivariate analysis.

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
