# Peer review of "COVID-19 and Inflammatory Bowel Disease: Patient Knowledge and Perceptions in a Single Center Survey"

_medicina, 2020, doi:10.3390/medicina56080407_

Round 1

Reviewer 1 Report

This is an interesting novel paper about the relationship of IBD patients (and their social/disease characteristics) and COVID pandemic. I have the following comments/corrections: 

repeated use of the expression "knowledge of COVID-19". i believe this is wrong, as you probably mean knowledge about COVID-19 or more properly knowledge about clinical importance of COVID-19. 

line 24 "one hundred twenty two(71%) were completed - of what??please add questionnaires and avoid or explain the percentage

line 73 -75 - please rephrase, concern is used twice

line 93 "clinic characteristics" please change it to clinical

line 101 "type" - please change it to "disease type"

line 101 "type of treatment" - please rephrase in order to avoid word type two times in one sentence

line 120-121 better explain types of anxiety - trait and state

table 1 - values are number(%)??? better explanation please

lines 172-174 I do not really understand the meaning. 

lines 242-244 meaning not clear

line 248 "routinely activities" please change to "routine activities"

line 249 "rearrangement of care" ???

line 255 " to the to the" - please correct

More explanations about the material sent to the patients and the fact that these patients had lower level of knowledge. The material was only sent to some patients? when they participated in the study was it clear if the received the material or if they had also studied it?

Thank you

Reviewer 2 Report

Thank you for submitting your valuable study to Medicina.

This study could be clinically significant because it can show the clinician the knowledge status patients. However, there are several things to be improved.

 1. Please describe about the patients permission for the enrollment of survey. Did you get the paper permission or oral permission? It is a ethical problem. So, please describe in the methods

2. Please show the OR of multivariate analysis in both Table 1 and 2.

3.Can you show me the items of each survey?

Reviewer 3 Report

Reviewer's report

Title:COVID-19 and Inflammatory Bowel Disease: patient knowledge and perceptions in a single center survey

Date: 31.07.2020

Reviewer's report:

The content of the subject "COVID-19 and Inflammatory Bowel Disease: patient knowledge and perceptions in a single center survey" has a value of interest and this is a well-written manuscript. I only have minor revisions to improve the manuscript.

Minor revisions:

  • Materials and methods: Line 90 needs a reference (... standard clinical, endoscopic, histological, and radiological criteria), Authors could use ECCO guideline as a reference if they like.
  • Results: I would recommend authors to restructure Table 1 as UC, CD and total. 
  • Discussion: Line 244 the references 6 and 7 are not formatted
  • Discussion: Authors mentioned some limitations but I would like to see a bit more than that maybe they could write a few sentences about the limitations of the study so it would be clear to everyone.
  • There are some places where abbreviations are not used consistently. I kindly recommend checking abbreviations.

Discretionary Revisions:

- I think that it is beneficial for researchers and clinicians to have the questionnaire that authors` have used available as a supplementary file. So that readers have a better idea about the full concept.

Reviewer 4 Report

This is an interesting study evaluating the knowledge of COVID-19 infection, disease management, prevention measures, and anxiety level among patients with Ulcerative Colitis and Crohn's disease in a single center for IBD in Italy, a country with a significant proportion of individuals affected by COVID-19.

There are no specific comments.

Author Response

No comment to reply